# Improved patient satisfaction and diagnostic accuracy in skin diseases with a Visual Clinical Decision Support System—A feasibility study with general practitioners

**Eckhard W. Breitbart[1], Kohelia Choudhury[1], Anders Daniel Andersen[2], Henriette Bunde[1], Marianne Breitbart[1], Antonia Maria Sideri[2], Susanne Fengler[1], John Robert Zibert⬡[2]***

**1** Association of Dermatological Prevention (ADP), Hamburg, Germany, **2** LEO Innovation Lab, Copenhagen, Denmark

* johnzibert@gmail.com

**Data Availability Statement:** All anonymised data underlying the results presented in the study is available in the manuscript and its Supporting

## Abstract

Patient satisfaction is an important indicator of health care quality, and it remains an important goal for optimal treatment outcomes to reduce the level of misdiagnoses and inappropriate or absent therapeutic actions. Digital support tools for differential diagnosis to assist clinicians in reaching the correct diagnosis may be helpful, but how the use of these affect patients is not clear. The primary objective of this feasibility study was to investigate patient experience and satisfaction in a primary care setting where general practitioners (GPs) used a visual clinical decision support system (CDSS) compared with standard consultations. Secondary objectives were diagnostic accuracy and length of consultation. Thirty-one patients with a dermatologist-confirmed skin diagnosis were allocated to consult GPs that had been randomized to conduct either standard consultations (SDR, n = 21) or CDSS (n = 16) on two separate study days one week apart. All patients were diagnosed independently by multiple GPs (n = 3–8) in both the SDR and CDSS study arms. Using the CDSS, more patients felt involved in the decision making (P = 0.05). In addition, more patients were exposed to images during the consultations (P = 6.8e-27), and 83% of those that were shown images replied they felt better supported in the consultation. The use of CDSS significantly improved the diagnostic accuracy (34%, P = 0.007), and did not increase the duration of the consultation (median 10 minutes in both arms). This study shows for the first time that compared with standard GP consultations, CDSS assist the GP on skin related diagnoses and improve patient satisfaction and diagnostic accuracy without impacting the duration of the consultations. This is likely to increase correct treatment choices, patient adherence, and overall result in better healthcare outcomes.

## Introduction

Patients are nowadays considered consumers of health services, and the assessment of their perspective has become a priority in medical management [1]. Furthermore, the shift to a

Information files. If further clarification is needed please contact corresponding author Dr John Zibert at johnzibert@gmail.com, or through Data Security Manager Kian Conteh at kian. conteh@leoilab.com.

**Funding:** The Study was funded by LEO Innovation Lab (sponsor) and provided salaries to the authors AMS, ADA and JRZ, and financial support to ADP to conduct the study. Sponsor was responsible for the study according to the law and regulations, and also had a role in study design, data collection and analysis, decision to publish and preparation of the manuscript.

**Competing interests:** EB, KC, HB, and SF declare no competing financial interests. AMS, ADA, and JRZ are employees of LEO Innovation Lab, this does not alter our adherence to PLOS ONE policies on sharing data and materials.

biopsychosocial model of healthcare and patient-centered care has emphasized the patient's part in medical decision making, and promoted more shared-decision making [2]. When patients participate in medical decision-making evidence suggests that they are more satisfied with their care, which can result in higher compliance to health regimens and better health outcomes [3–5]. Patient satisfaction is therefore a common indicator for quality healthcare. Other important aspects of quality healthcare may be diagnostic accuracy and length of consultations in order to achieve clinical effectiveness and patient safety [6, 7]. To this end, support tools for better differential diagnoses may be helpful.

A visual clinical decision support system (CDSS) may improve diagnostic accuracy of skin conditions and reduce misdiagnosis-related harms. A CDSS is a computer-assisted differential diagnosis tool with content and potentially images of medical conditions [8]. Often it can be accessed using computers, tablets and smartphones (via websites and applications). Unlike textbooks indexed by disease, CDSS allows doctors to enter patient demographics like age, sex, and objective and subjective symptoms, including lab-findings. Based on this information the CDSS may suggest the GP a diagnosis that matches the entered criteria, enabling doctors to "rule in or out" by comparing a patient's skin condition with the information and potentially images given to match the condition. Often for each differential the CDSS also provides concise content on disease-specific information as well as information on management, therapy, and handouts for patients which may be used to improve the patient-doctor communication.

In a randomized controlled trial (RCT) the use of CDSS by non-dermatologists improved the diagnostic accuracy of skin complaints compared with textbook use [9], and in another study CDSS was shown to assist GPs in correctly diagnosing cellulitis, a commonly misdiagnosed condition [10]. To our knowledge, no studies have previously investigated how CDSS can impact diagnostic accuracy in a group of patients with a wider range of more complex skin diseases. Diagnostic accuracy is usually the main focus in studies with CDSS, whereas the patient's perspective is often not investigated. To our knowledge, despite the widespread use of CDSS there is no well-documented information on how its use affects patient experience and satisfaction and relates to diagnostic accuracy. We conducted this feasibility study [11] to assess how CDSS vs standard (SDR) consultations affect patient satisfaction, diagnostic accuracy and length of consultations.

## Material and methods

### Study design and population

The study was a randomized feasibility study conducted in a dermatology clinic at Hautarztpraxis Buxtehude, Buxtehude, Germany over two study days in November 2016. Ethical approval for this study was obtained from the ethics committee of the Ärztekammer Niedersachsen, Germany. Following verbal information given by the dermatologists, written consent was obtained from each test patient prior to study entry.

On the first study day, all consultations using the visual CDSS application (VisualDx, Rochester, NY, USA) available on a tablet (iPad, Apple, CA, USA) were conducted and the following week the standard (SDR) consultations were conducted, in which a tablet (iPad, Apple, CA, USA) to access the internet (Safari, Apple, CA, USA) was made available. The study was designed so that test patients (n = 31) were examined independently by multiple GPs first in the CDSS arm and then in the SDR arm. Patients were males and females over 18 years of age resident in Germany recruited by two dermatologists from their own pool of patients. All patients had chronic and/or newly diagnosed skin diseases that had been confirmed by two independent senior dermatologists (Golden Standard) at ADP, Buxtehude. To ensure a mix of complexity of the cases, a broad range of skin conditions was included. This was important for

the study to best reflect a realistic range of skin diseases that GPs can be exposed to in a primary care setting. The patient cases included common skin diseases such as basal cell carcinoma, psoriasis and eczema, and also more uncommon diseases, that practicing GPs are still likely to encounter in-office, such as trichoepithelioma, morphea and granuloma faciale.

Recruitment of GPs was carried out in cooperation with the Bezirks Landesärztekammer Stade (Regional Medical Association of Stade, Germany). The GPs, randomized to the CDSS arm and control arm, respectively, were all practice-based English-speaking GPs that as a minimum had completed their residency, and were accustomed to the use of internet and mobile technology. The GPs in the CDSS arm were offered to attend a CDSS webinar (75% attended), and also had the opportunity to further familiarize themselves with the tool in the two weeks prior to the study day. GPs with previous experience with electronic differential builders, extra education and/or special interests in dermatology were excluded.

## Study assessments

Patient satisfaction in both the CDSS arm (n = 159) and SDR arm (n = 175) was assessed after each examination by a list of 11 questions regarding their experience with the consultation (S1 Table). The first six questions were reflecting patients' general satisfaction level, to which they should assign a score from 1 (poor) to 5 (excellent), and were constructed to assess how they had experienced the interpersonal communication with the GP during the consultation. The following five were related to the patient's observations of the GP's behavior and actions. Questions 1–6 were taken from a validated communication assessment tool developed to assess how patients experience physicians' communication skills [12]. The remaining questions were specifically related to the current study and consequently developed for the purpose of this study. Electronic case report forms (CRFs) were filled out by the GP during the consultation and used to evaluate the diagnostic accuracy, suggested next steps and consultation lengths (S2 Table). Consultation lengths were originally expected to last approximately 15 minutes. Consultations lasting either less than 2 minutes or more than 20 minutes (n = 6 CDSS, n = 5 SDR) were considered outliers and excluded from the analyses. Questionnaires and CRFs were filled either on paper or online (Google Forms, CA, USA).

## Statistical analysis

All figures and statistical analyses were performed in Python version 2.7.12 and scipy.stats package. Study results were summarized into tables and figures where appropriate, and normality of data was checked with the Shapiro-Wilk test and qq-plots.

Demographics were analyzed with Mann Whitney U test and $\chi^2$. Patient satisfaction questionnaires were divided into patient-specific questions (Q1-Q6) and those that were consultation specific (Q7-Q10). The former was treated as paired samples (n = 31) and analyzed using the Wilcoxon signed-rank test, in which the overall median score across the six questions and the individual scores per question per patient were compared between study arms. Finally, also the proportion of excellent scores (score 5) given by each patient in each study arm was compared.

The latter questions (Q7-Q10) with binary outcomes were treated as independent samples and compared between consultations in each study arm (n = 334) using $\chi^2$. Also, the consultation length and diagnostic accuracy were treated as independent samples and compared between groups using Mann Whitney U test and $\chi^2$, respectively. Correlations between the overall satisfaction (Q1-Q6), and the GPs age, years of practice and the level of practice with the CDSS prior to study day in the CDDS group were investigated using Pearson correlation coefficient. Finally, also the influence of GPs age, years of practice and length of consultations

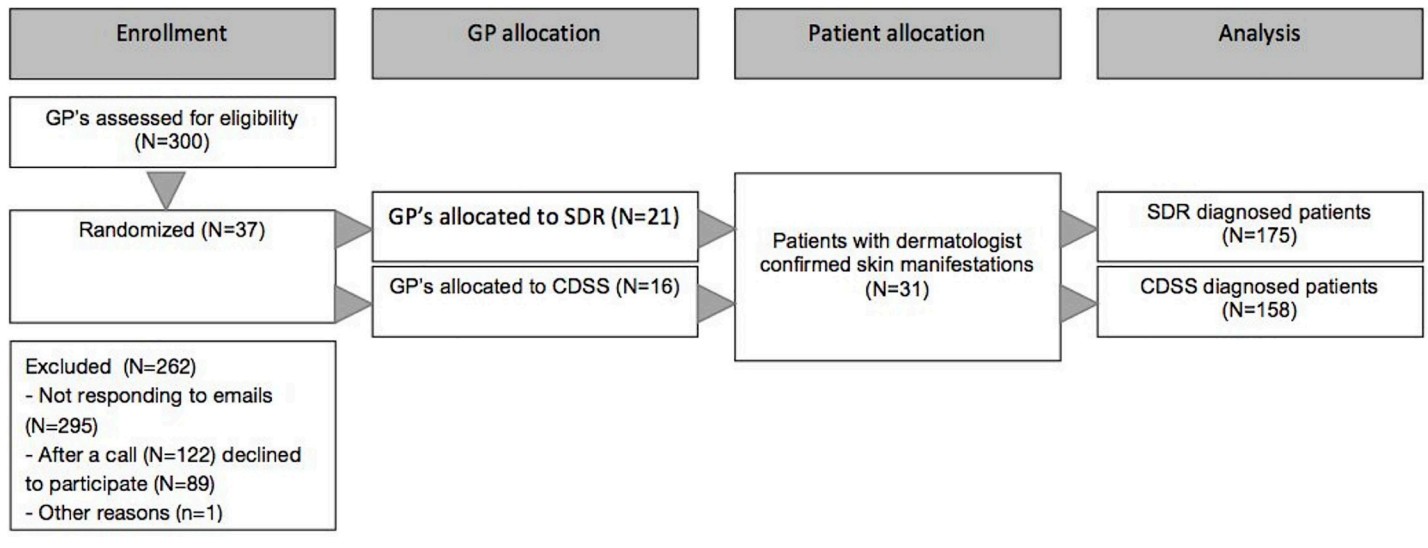

**Fig 1. CONSORT flow diagram.**

on the diagnostic accuracy was investigated by comparing these variables between groups of correct and incorrect diagnoses, respectively. This was performed within consultations in each study arm, and across all consultations (n = 334) using the Mann Whitney U test.

## Results

In total 31 patients participated in the study (Fig 1). Test patients had a median age of 61 (21–79) years, and 21 were males whereas 10 were females. The patients altogether represented 21 different diagnoses (Fig 2). Among the 37 GPs that participated in the study, the median age was 46 (38–52), with a median of 12 (8–20) years of experience. Although GPs were initially randomized to either the CDSS or SDR arm, a few were unavailable on the scheduled study day, and were therefore transferred to the other study arm based on their availability. The GPs were not aware of which study day was the CDSS or the SDR day, respectively, and this was therefore not a part of such a decision. This resulted in more GPs being represented in the SDR arm. The same pool of patients was available on the two study days. By chance, GPs in the CDSS-arm compared with SDR-arm were older and more experienced (Table 1).

This was a feasibility study, single-center, open-label, within-patient comparison to explore if the use of a visual clinical decision support system (CDSS) compared with standard consultation (SDR) among GPs impacted the patient experience and satisfaction with the consultation, the diagnostic accuracy and the consultation length.

A total of 334 consultations were conducted. This resulted in 159 and 175 patient satisfaction questionnaires, and diagnoses in the CDSS and SDR arms, respectively. The median number of consultations per patient in the two groups was comparable being 6 for CDSS and 5 for SDR (P = 0.12). In 75 of the 175 consultations in the SDR arm, the GPs reported to use supporting tools such as: internet on tablet (n = 51), textbooks (n = 16), textbooks and internet on tablet (n = 5); and dermatoscope (n = 3).

### Patient satisfaction with the consultation

After each consultation in either the SDR or CDSS arm the patients completed a questionnaire consisting of 11 questions regarding their experiences with the GP during the consultation (S1

Table). Significantly more patients from the consultations with CDSS reported to fully agree (gave the maximum score) with the statement that '*the doctor involved me in decisions as much as I wanted*' compared with a standard consultation (P = 0.05, Fig 3A). The questions related

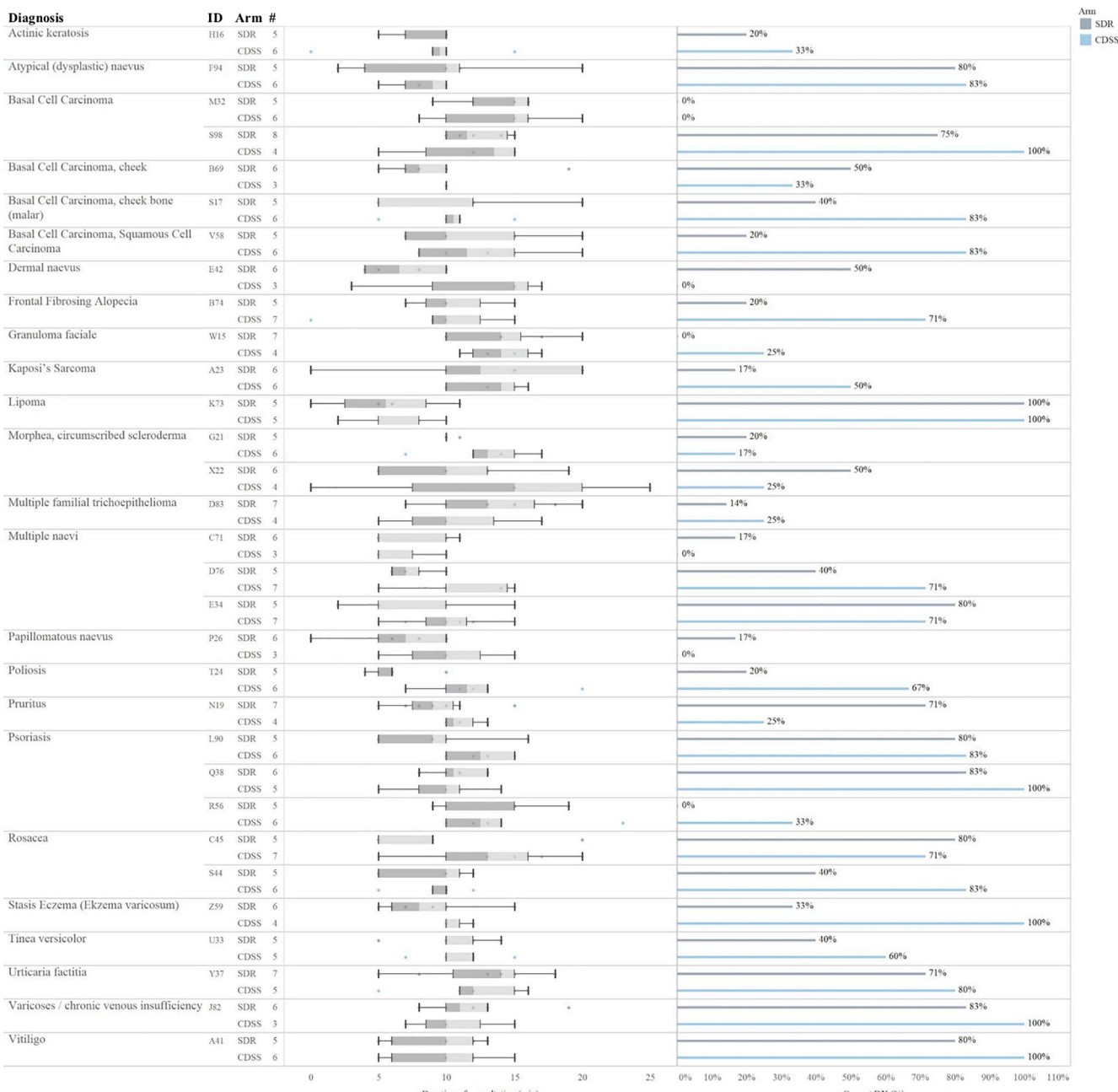

**Fig 2. In the majority of consultations CDSS improved the accuracy of the diagnostic outcome.** In total there were 21 different diagnoses. In 65% of the individual cases CDSS resulted in a more correct diagnosis than SDR (20/31 for CDSS, 9/31 for SDR). The following diagnoses received a *high correctness in both consultation types* (CDSS/SDR): atypical pigmented nävi (83/80), chronic venous insufficiency (100/83), lipoma (100/100), multiple nävi (71/80), psoriasis (83/80), rosacea (71/80), urticaria (80/71), vitiligo (100/80). For eight diagnoses of *both consultations type the correctness was poor*, that being: actinic keratoses (33/20), BCC (0/0), granuloma faciale (25/0), morphea (17/20, 25/50), multiple nävi (0/17), psoriasis (33/0). The *largest difference between the two groups* was found in 10 diagnoses: BCC (83/40, 83/20), alopecia (71/20), hereditary kaposi's sarcoma (50/17), morphea (25/50), dermal nevus (0/50), poliosis (67/20), pruritus (25/71), psoriasis (100/71, 33/0), rosacea (83/40), stasis dermatitis [Dermatitis varicosa] (100/33). The consultation duration varied across the different diagnoses as well as the diagnostic accuracy per patient. Consultations lasting less than 2 minutes or more than 20 minutes were considered outliers and excluded from analyses.

Table 1. Characteristics of the participating GPs in the two study arms.

|  | CDSS n = 16 | SDR n = 21 | P-value |
|---|---|---|---|
| Age* | 48 (40–52) | 41 (38–49) | 9.4e-05 |
| Gender, *n* (m/f) | 9/7 | 7/14 | 0.289 |
| Years of experience | 16 (11–20) | 11 (8–19) | 0.00028 |

*Median (25th-75th). Mann Whitney U test, $\chi2$

to patient's satisfaction with the consultation (question 1–6) were similar between the two arms (P = 0.83), and for the individual questions there were also no significant differences between the mean scores in the two study arms.

In the SDR arm in the current study the internet was frequently used (S1 Appendix), which is oftentimes the case in conventional consultations at the GP's office. In this study more than 80% of the patients reported that they were not bothered by the use of textbooks, the internet or CDSS on a tablet during the consultation, and this was not different between the CDSS and the SDR arms (P = 0.12) (Fig 3B). Showing images as part of the consultation was much more frequent in the CDSS (70%) compared with the SDR (13%) consultations (Fig 3B, P = 6.8e-27). Overall 83% of the patients that were shown images during the consultation reported that they felt better supported by this.

## Diagnostic accuracy and duration of consultations with CDSS

Overall, the diagnostic accuracy improved relatively with 34% in the CDSS compared to the SDR study arm (P = 0.007, Fig 4). Also, in the CDSS relative to the SDR arm the patients more frequently (+22%) replied that the diagnosis they had been given by the GP at the end of the consultation matched that provided by the dermatologist prior to the study (Fig 3B, P = 0.034). By chance, some differences in the characteristics of the participating GPs existed (Table 1), however, the diagnostic accuracy was not affected by the GPs age, years of medical practice, level of practice with the visual CDDS, or the length of consultation (S3 Table).

In total, the 31 test patients represented 21 different diagnoses (Fig 2). For each individual diagnosis among the test patients the following observations regarding the correctness of the diagnosis given in the two different study arms were made. A high level of correct diagnoses in both consultation types (CDSS, SDR) were observed for atypical (dysplastic) naevus (83%, 80%), varicoses/chronic venous insufficiency (100%, 83%), lipoma (100%, 100%), multiple naevi (71%, 80%), psoriasis (83%, 80%), rosacea (71%, 80%), urticaria factitia (80%, 71%), and vitiligo (100%, 80%). In contrast, for the following eight diagnoses the correctness of the diagnosis given was poor for both consultation types: actinic keratosis (33%, 20%), basal cell carcinoma (0%, 0%), granuloma faciale (25%, 0%), morphea/circumscribed scleroderma (17%, 20%; 25%, 50%), multiple naevi (0%, 17%), psoriasis (33%, 0%). The largest difference in the correctness of the diagnosis given during the consultation between the two study arms was found in the following 10 diagnoses: Basal cell carcinoma (83%, 40%; 83%, 20%), frontal fibrosing alopecia (71%, 20%), kaposi's sarcoma (50%, 17%), morphea/circumscribed scleroderma (25%, 50%), dermal naevus (0%, 50%), poliosis (67%, 20%), pruritus (25%, 71%), psoriasis (100%, 71%; 33%, 0%), rosacea (83%, 40%), stasis dermatitis (dermatitis varicosa) (100%, 33%) (Fig 2). Overall, there was no pattern in which diagnoses were wrongfully diagnosed by the GPs in the study (S1 Appendix). In general, GPs using the CDSS were less inclined to want to refer the patient to a dermatologist following the consultation than GPs in the SDR arm. As such, in the consultations in which correct diagnoses had been given 62.2% of GPs in the CDDS arm compared with 75% of GPs in the SDR arm intended to refer the patient to a

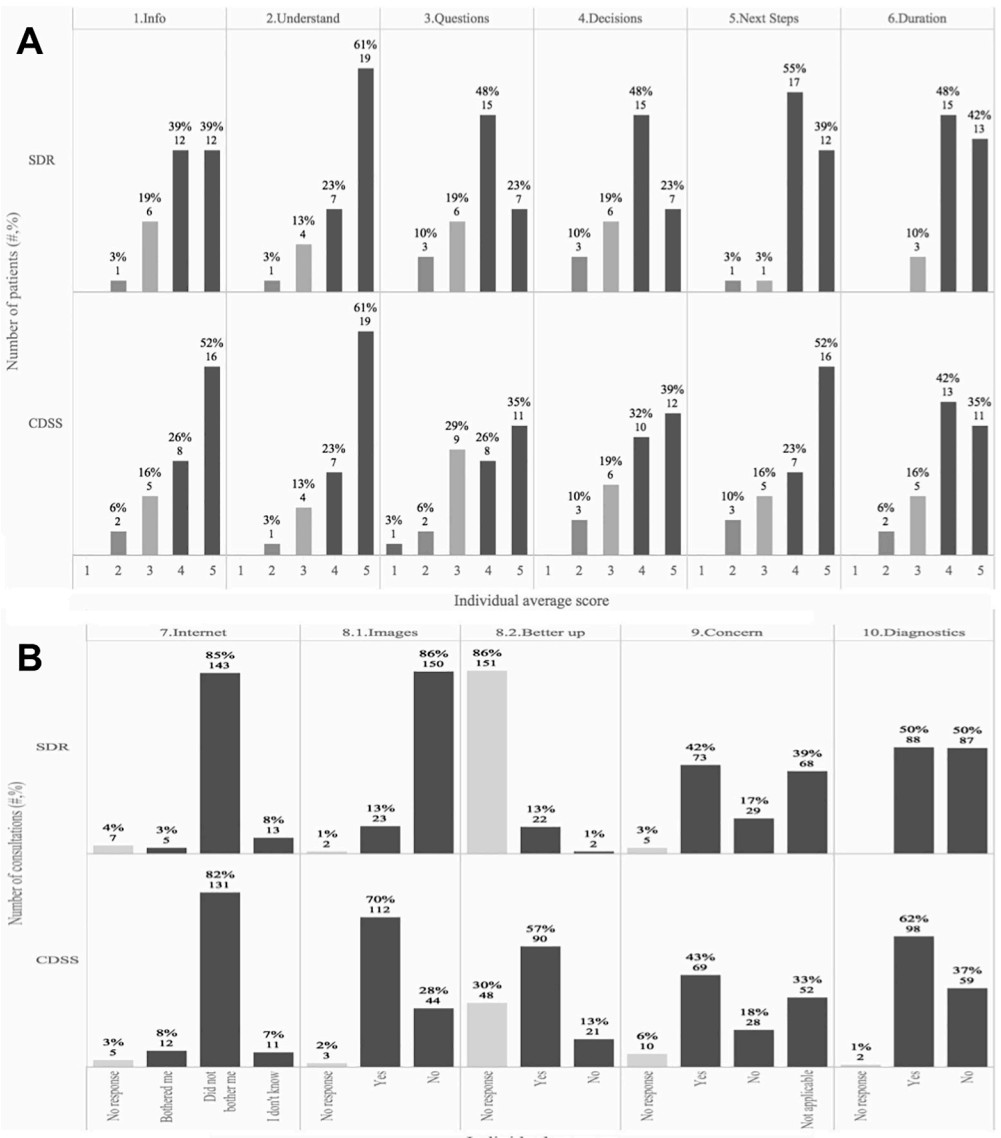

**Fig 3. CDSS improves the patient satisfaction.** After each consultation in either the SDR (n = 175) or CDSS arm (n = 159) the patients were asked 11 questions regarding their experience with the GP (S1 Table). (A) Illustration of the first 6 questions. The overall mean across all questions was not different. For the individual questions there was also no significant difference between the means. In the CDSS compared with the SDR consultations, significantly more patients gave the maximum score (score 5) in question 4 (decision), P = 0.05. (B) Illustration of the last 5 questions. For question: 7) patients were generally not bothered that the GP used the internet or CDSS on a tablet or textbook, and there were no differences between study arms, 8.1) patients experienced that the GP used images with CDSS to a much higher degree (P = 6.78e-27), 8.2) patients felt better supported when the GP used images; 10) and for the CDSS relative to the SDR patients did the diagnose given by the GP to a higher degree match the diagnosis that the patient was given by a dermatologist prior to the study (P = 0.034).

specialist (P = 0.087). Similarly, in the consultations that ended with GPs giving the wrong diagnosis, fewer GPs in the CDSS arm (64.1%) than in the SDR arm (74.3%) intended to refer the patients (P = 0.057). In the study, reporting the ICD-10 codes after diagnosing the patient was not mandatory. However, GPs using the CDSS more often reported the ICD-10 codes inasmuch as 51% of all the consultations compared with only 18% of the consultations in the

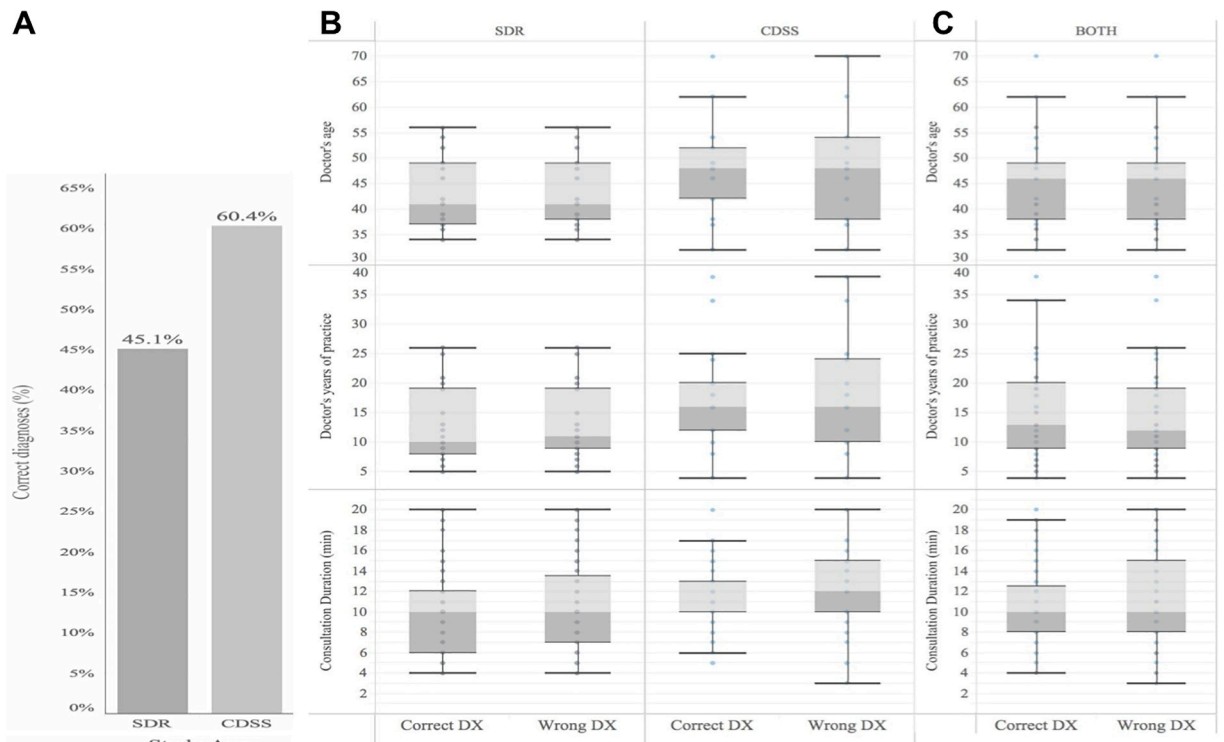

**Fig 4. Diagnostic accuracy significantly improved with CDSS compared to SDR.** In total n = 175 and n = 159 independent consultations with the diagnosis and patient information as the outcome were carried out for SDR and CDSS, respectively. (A) The diagnostic correctness as compared with the golden standard of the diagnosis from two independent senior dermatologists, was significantly higher with CDSS compared with SDR with a relative difference of 34% (P = 0.006). (B). Whisker plot for GPs who used CDSS or SDR, and who diagnosed correctly or not with respect to GPs age, experience and length of consultation (no significant differences). (C) Combined for both CDSS and SDR (no significant differences).

SDR arm the ICD-10 codes were recorded. Generally, when a GP made a correct diagnosis, the ICD-10 code was correct in 89.5% (51/57) and 86.6% (13/15) cases in the CDSS and SDR arm, respectively.

The median duration of the consultations using the CDSS was 10 minutes, which was similar to that in the SDR arm. In the CDSS arm the overall patient satisfaction (questions 1–6) correlated negatively with increased duration of consultation (P = 0.02). Across both study arms younger patients (20-40y) tended to be more bothered about consultation length relative to the older test patients (P = 0.069). Information on the consultation lengths per patient per study arm can be seen in Fig 2.

## Discussion

Patient satisfaction is an important part of today's healthcare. Involving patients in the medical decision-making increases patient satisfaction, and may lead to increased adherence to treatments and better health outcomes [3–5]. Also, diagnostic accuracy and the time spent on consultations are important aspects. The primary objective of this feasibility study was to investigate if CDSS would affect patient experience and satisfaction in a cohort of patients with dermatologist-confirmed skin diagnoses in a primary care setting. Secondary objectives were to directly compare the diagnostic accuracy and consultation lengths of CDSS consultations with standard medical consultations. Overall, we show that the CDSS increased aspects of

patient satisfaction, improved diagnostic accuracy without influencing the duration of the consultation.

In this study we show for the first time that when GPs use the CDSS more patients feel sufficiently involved in the medical decisions during the consultation. The five-point poor-excellent scale that was used has previously been shown to be best in tests for this purpose [12]. We found that 39% of the patients in the CDSS arm compared with only 23% in the SDR arm reported that the doctor did an excellent job involving them in the medical decision making, but we did not observe group differences based on overall means. Correspondingly, Makoul *et al.* previously concluded that it was more meaningful to dichotomize the scores by reporting the proportion of "excellent" ratings rather than by summarizing scores via patient-reported means. Our findings clearly show some advantages of the CDSS tool. Very importantly, our data also suggest that patients generally are not bothered by the GPs using the internet or the CDSS on a tablet or a textbook during the consultation, which obviously is a prerequisite for successful implementation of such a tool with GPs in the future. However, there was a trend that the younger patients felt more bothered with this. This could partly be explained by the fact that this generation, and millennials overall, are typically savvier using digital solutions, and consequently may be more intolerant towards inexperienced and slow users. This could also partly explain why overall satisfaction with the consultation correlated negatively with the length of the consultation which was only evident in the CDSS arm where the GPs were using a tablet.

It is generally known that visual cues, such as the use of images to support explanations, for many people have a higher didactic value. In this study, GPs using the CDSS generally showed more images to help explain the patient's situation, which is an inherent part of the tool to help improve the patient experience and education. The value of this was also confirmed in our study where patients reported to feel better supported by this.

The 34% increase in diagnostic accuracy by GPs using the CDSS is another key finding in the study, which was demonstrated in spite of the fact that the participating GPs were relatively inexperienced in using it. This is in agreement with previous studies in which improved diagnostic accuracy has been demonstrated among non-dermatologists in skin-related conditions [9, 10]. In one RCT the performance of the tool was compared with a textbook use [9], and in another it aided in correctly diagnosing commonly misdiagnosed conditions like cellulitis [10]. In this feasibility study, the different diagnoses selected were a mix of both easy and difficult cases to diagnose. This was designed to reflect a realistic list of conditions that a GP may encounter at the office over a 6-month period. This study showed that some skin diseases were generally difficult for GPs to diagnose. Conditions that were more difficult to diagnose comprised very common skin conditions such as actinic keratosis, basal cell carcinoma, multiple naevi, psoriasis, rosacea, and stasis eczema as well as rare skin diseases like frontal fibrosing alopecia, Kaposi sarcoma, morphea, and poliosis. The improvement in diagnostic accuracy from 45% to 60% is high, and equal to a 34% relative increase. However, it also highlights a need for diagnostic tools and further education for GPs to help them better diagnose skin-related conditions correctly. This could, in turn, optimize treatment outcomes, likely to be linked to optimized treatment plans and increased adherence. All parameters that from a healthcare cost perspective will be beneficial.

Clinical images of the common diagnoses can vary considerably and misdiagnosis by GPs and even by experienced dermatologists can occur. For this reason, a visual clinical decision support system with a wide range of clinical images of common and rare dermatological diagnosis can be helpful. It provides a very good opportunity to improve the diagnostic accuracy, which is likely to increase the possibility that the correct treatment is being given. This arguably will influence patient satisfaction and have a positive impact on the healthcare system.

The study has limitations that should also be considered when interpreting the implications of our findings. The study was designed so that patients were first consulted by GPs in the SDR arm, and then by the GPs using the CDSS. This may have influenced how patients perceive the consultation and thereby respond to the questionnaires. Also, another limitation is that the participating GPs were not fully randomized to their respective study-arm inasmuch as some GPs were scheduled for the specific study day according to actual availability. Furthermore, this study only reflects one CDSS and other systems may result in different outcomes.

Because of the general lack of dermatologists, it can be speculated that providing GPs with a CDSS may improve treatment outcomes for patients and reduce the number of referrals, which at large may be beneficial from a health-economic perspective. It was interesting to note that by using the CDSS more GPs chose not to refer the patient, irrespective of whether they had given the right or wrong diagnosis. For most skin conditions, when diagnosed correctly in primary care, the GPs are also able to prescribe and initiate the correct treatment plan, which can reduce the number of referrals to dermatologists. Overall, referrals to a dermatologist within the EU countries are generally costly. A reduction in the number of referrals could indicate a health care saving equivalent to the reimbursement of a referral, which in Germany is a cost saving of 14 EUR/visit (2017 benchmark). However, it should be mentioned that the overall cost-effectiveness of implementing different eHEALTH-solutions in the healthcare system is still being debated [13]. Nonetheless, in addition to potential health care savings, being diagnosed correctly by a GP is likely to bring other benefits. Arguably, this would improve the patient experience, and potentially also the patients' satisfaction with and trust in the healthcare system.

## Conclusions

In this study the use of visual CDSS in a primary care setting increased patient satisfaction with the consultations with respect to decision making, better understanding of their condition because of the GPs use of images, and the correctness of the diagnosis. The diagnostic accuracy increased relatively with 34% with the use of CDSS compared to standard consultations. These results show that if implemented in the GP's office, CDSS may effectively support improving patient outcomes, and increase diagnostic accuracy. This could be likely to reduce healthcare spending significantly.

## Supporting information

**S1 Checklist.**
(DOC)

**S1 Appendix. Overview of wrong diagnoses.** Overall, the pattern was very heterogeneous using both CDSS or SDR.
(TIF)

**S2 Appendix. Internet activity of the participating doctors.** During each consultation the internet activity was registered. The most frequently used websites in both groups were Google search engine, Wikipedia and Google books. For one GP in the CDSS-arm the browsing history on internet use was not retrievable because the GP used the CDSS application.
(TIF)

**S1 Table. Patient satisfaction questionnaire.**
(DOCX)

**S2 Table. GP questionnaire.** *Required. CDSS: clinical decision support system (n = 16 GPs), SDR: standard (n = 21 GPs).
(DOCX)

**S3 Table. P-values* reflecting the influence of GP demographics, length of consultation and practice with the CDSS before the study day (CDSS group only) on diagnostic correctness, in the SDR arm, the CDSS arm, and across both arms, respectively.** *P-values were calculated using Mann Whitney U test comparing those giving the right diagnosis with those giving the wrong diagnosis.
(DOCX)

**S1 File.**
(PDF)

# Acknowledgments

We thank Veronika Deichmann and Kathrin Fuhrmann for the recruitment of study subjects and for assistance with the study administration, Art Papier and Ann Komanecky for conducting the CDSS webinars, and Paul Jacobs for providing assistance during the study conduction.

# Author Contributions

**Conceptualization:** Eckhard W. Breitbart, Kohelia Choudhury, Anders Daniel Andersen, Susanne Fengler, John Robert Zibert.

**Data curation:** Eckhard W. Breitbart, Kohelia Choudhury, Anders Daniel Andersen, Antonia Maria Sideri, Susanne Fengler, John Robert Zibert.

**Formal analysis:** Eckhard W. Breitbart, Kohelia Choudhury, Anders Daniel Andersen, John Robert Zibert.

**Funding acquisition:** Eckhard W. Breitbart, Kohelia Choudhury, John Robert Zibert.

**Investigation:** Eckhard W. Breitbart, Kohelia Choudhury, Henriette Bunde, Marianne Breitbart, Susanne Fengler.

**Methodology:** Eckhard W. Breitbart, Kohelia Choudhury, Anders Daniel Andersen, Antonia Maria Sideri, Susanne Fengler, John Robert Zibert.

**Project administration:** Kohelia Choudhury, Anders Daniel Andersen, Henriette Bunde, Susanne Fengler, John Robert Zibert.

**Resources:** Kohelia Choudhury, Anders Daniel Andersen, Henriette Bunde, Susanne Fengler, John Robert Zibert.

**Software:** Anders Daniel Andersen, Antonia Maria Sideri, John Robert Zibert.

**Supervision:** Eckhard W. Breitbart, Anders Daniel Andersen, Susanne Fengler, John Robert Zibert.

**Validation:** Eckhard W. Breitbart, Kohelia Choudhury, Anders Daniel Andersen, Antonia Maria Sideri, John Robert Zibert.

**Visualization:** Kohelia Choudhury, Anders Daniel Andersen, Antonia Maria Sideri, John Robert Zibert.

**Writing – original draft:** Eckhard W. Breitbart, Kohelia Choudhury, Anders Daniel Andersen, Henriette Bunde, Marianne Breitbart, Antonia Maria Sideri, Susanne Fengler, John Robert Zibert.

**Writing – review & editing:** Eckhard W. Breitbart, Kohelia Choudhury, Anders Daniel Andersen, Henriette Bunde, Marianne Breitbart, Antonia Maria Sideri, Susanne Fengler, John Robert Zibert.

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
