## [Decision Letter · Decision Letter 0]

3 Mar 2020

PONE-D-19-34156

Improved patient satisfaction and diagnostic accuracy in skin diseases with a Visual Clinical Decision Support System - a feasibility study with general practitioners

PLOS ONE

Dear Dr Zibert,

Thank you for submitting your manuscript to PLOS ONE. After careful consideration, we feel that it has merit but does not fully meet PLOS ONE’s publication criteria as it currently stands. Therefore, we invite you to submit a revised version of the manuscript that addresses the points raised during the review process.

We would appreciate receiving your revised manuscript by Apr 17 2020 11:59PM. To enhance the reproducibility of your results, we recommend that if applicable you deposit your laboratory protocols in protocols.io, where a protocol can be assigned its own identifier (DOI) such that it can be cited independently in the future. For instructions see: http://journals.plos.org/plosone/s/submission-guidelines#loc-laboratory-protocols

We look forward to receiving your revised manuscript.

Kind regards,

Elisa J. F. Houwink, MD, PhD

Academic Editor

PLOS ONE

Additional Editor Comments (if provided):

Before considering publication of your work, major revisions should be made.

See below:

This study is about patients’ perspectives on the use of CDSS (VisualDx) by GPs on diagnosing dermatologic condition compared to standard consultation. The authors claim that the use of CDSS improves patient satisfaction, improves diagnostic accuracy and reduces referrals to specialists.

While the study explores noble concept, the language is unclear in some parts, making it difficult to follow. I advise the authors to work on improving the flow and readability of the text.

1. Foremost, the study does not appear to be sound. The study design is questionable and may not measure what the authors are trying to do. The patients knew their diagnosis prior to being evaluated by either CDSS or SDR, which may affect how they perceive the visits/satisfaction score.

2. The authors mention that the objectives of the study are to assess patient satisfaction, diagnostic accuracy and length of consultation, but in conclusion (especially in abstract), they mention patient satisfaction, diagnostic accuracy and number of referrals. I recommend that the authors answer the questions initially intended to answer.

3. The authors say, “Other important aspects of quality healthcare are diagnostic accuracy and length of consultations (line 56).” Please include the evidence that supports this statement.

4. The authors mention 2 studies that showed the diagnostic accuracy of CDSS, so it is unclear what the study’s objective on diagnostic accuracy adds to existing literature.

5. It is unclear how the consent process was taken place. I suggest that the authors include details of the recruitment process of the GPs. Was there any compensation for the participants (patients, GPs, dermatologist)? Also it is unclear if the patients were seen at GPs practice or at a different office, maybe in a research lab?

6. The authors mention that the GPs were randomized to SDR vs CDSS, but based on their personal availability. This does not seem to fit the randomization process. Also unclear if the GPs knew which day was CDSS day and which day was SDR day prior to selecting the study day.

7. On line 113, the authors mention that the first 6 questions test patient’s actual satisfaction, but per the reference 10, the questionnaire measures patient view of physician communication skills, which is not synonymous to patient satisfaction.

8. On line 231, the authors mention, “the relative difference between the patient’s understanding of their diagnosis and the one provided during the consultation was 22% higher in the CDSS….” The statement is unclear as written, and makes it sound like there were more diagnostic discrepancies in CDSS arm. Also the figure number should be 3B instead of 2B.

9. The authors claim that there were no statistically significant differences among GPs in CDSS vs SDR on diagnostic accuracy. It is unclear if that is the case in each dermatologic condition since only 3-8 GPs evaluated each patient, not all GPs. Subgroup sample size may be too small to support that the diagnostic accuracy was due to CDSS use.

10. The authors mention tablet (in method section), internet, and iPad (results and discussion sections). It is unclear if they all mean the same thing for the purpose of this study. Did the GPs have access to internet via some other means other than a tablet/iPad? Did GPs in SDR arm use tablet/iPad for any other use besides internet access?

11. The statement on line 291 says, “We found that 52% of the patients in the CDSS arm compared with only 39% in the SDR…..” However, according to figure, it is 39% and 23%, respectively.

12. On line 302, the authors argue that the negative correlation between overall satisfaction with the consultation and the length of the consultation was only evident in the CDSS arm where the GPs were using an iPAD. It is unclear if the GPs in SDR arm were also using iPad or not. In method section, the authors mention that both groups were provided with a tablet.

13. The last paragraph of the discussion section, the authors makes comments that are not the main objective of the study or supported by other studies. For instance, the authors say that CDSS can help achieve patient outcome and save money for healthcare system, but based on the most recent systemic review, there was no clear evidence that CDSS improves patient outcome or is cost effectiveness. (Black AD, Car J, Pagliari C, et al. The impact of eHealth on the quality and safety of health care: a systematic overview. PLoS Med. 2011;8(1):e1000387. Published 2011 Jan 18. doi:10.1371/journal.pmed.1000387

14. On line 335, the authors mention that the GPs in CDSS arm felt more confident with the diagnosis that they chose not to refer the patients. The questionnaire did not explore the GPs perspectives on how confident they felt, so we cannot jump to conclusion that they felt more confident based on the number of referrals placed. The reason for referral could be diagnostic, but also therapeutic.

15. The authors’ last comment on the discussion section (increased quality of life, happiness, increased productivity of the society) is too broad, and does not seem immediately relevant to the study objectives nor supported by the study.

16. The authors should include the limitation of the study (such as not being a randomized controlled trial, etc) to keep in mind the generalizability of the study.

Journal Requirements:

2) Please provide additional details regarding participant consent. In the ethics statement in the Methods and online submission information, please ensure that you have specified whether consent was suitably informed.

3) We note that you have indicated that data from this study are available upon request. PLOS only allows data to be available upon request if there are legal or ethical restrictions on sharing data publicly. For information on unacceptable data access restrictions, please see http://journals.plos.org/plosone/s/data-availability#loc-unacceptable-data-access-restrictions.

4) Please include captions for your Supporting Information files at the end of your manuscript, and update any in-text citations to match accordingly. Please see our Supporting Information guidelines for more information: http://journals.plos.org/plosone/s/supporting-information.

Reviewers' comments:

Reviewer's Responses to Questions

**Comments to the Author**

1. Is the manuscript technically sound, and do the data support the conclusions?

Reviewer #1: Partly

2. Has the statistical analysis been performed appropriately and rigorously? 

Reviewer #1: I Don't Know

3. Have the authors made all data underlying the findings in their manuscript fully available?

Reviewer #1: Yes

4. Is the manuscript presented in an intelligible fashion and written in standard English?

Reviewer #1: No

5. Review Comments to the Author

Reviewer #1: This study is about patients’ perspectives on the use of CDSS (VisualDx) by GPs on diagnosing dermatologic condition compared to standard consultation. The authors claim that the use of CDSS improves patient satisfaction, improves diagnostic accuracy and reduces referrals to specialists.

While the study explores noble concept, the language is unclear in some parts, making it difficult to follow. I advise the authors to work on improving the flow and readability of the text.

1. Foremost, the study does not appear to be sound. The study design is questionable and may not measure what the authors are trying to do. The patients knew their diagnosis prior to being evaluated by either CDSS or SDR, which may affect how they perceive the visits/satisfaction score.

2. The authors mention that the objectives of the study are to assess patient satisfaction, diagnostic accuracy and length of consultation, but in conclusion (especially in abstract), they mention patient satisfaction, diagnostic accuracy and number of referrals. I recommend that the authors answer the questions initially intended to answer.

3. The authors say, “Other important aspects of quality healthcare are diagnostic accuracy and length of consultations (line 56).” Please include the evidence that supports this statement.

4. The authors mention 2 studies that showed the diagnostic accuracy of CDSS, so it is unclear what the study’s objective on diagnostic accuracy adds to existing literature.

5. It is unclear how the consent process was taken place. I suggest that the authors include details of the recruitment process of the GPs. Was there any compensation for the participants (patients, GPs, dermatologist)? Also it is unclear if the patients were seen at GPs practice or at a different office, maybe in a research lab?

6. The authors mention that the GPs were randomized to SDR vs CDSS, but based on their personal availability. This does not seem to fit the randomization process. Also unclear if the GPs knew which day was CDSS day and which day was SDR day prior to selecting the study day.

7. On line 113, the authors mention that the first 6 questions test patient’s actual satisfaction, but per the reference 10, the questionnaire measures patient view of physician communication skills, which is not synonymous to patient satisfaction.

8. On line 231, the authors mention, “the relative difference between the patient’s understanding of their diagnosis and the one provided during the consultation was 22% higher in the CDSS….” The statement is unclear as written, and makes it sound like there were more diagnostic discrepancies in CDSS arm. Also the figure number should be 3B instead of 2B.

9. The authors claim that there were no statistically significant differences among GPs in CDSS vs SDR on diagnostic accuracy. It is unclear if that is the case in each dermatologic condition since only 3-8 GPs evaluated each patient, not all GPs. Subgroup sample size may be too small to support that the diagnostic accuracy was due to CDSS use.

10. The authors mention tablet (in method section), internet, and iPad (results and discussion sections). It is unclear if they all mean the same thing for the purpose of this study. Did the GPs have access to internet via some other means other than a tablet/iPad? Did GPs in SDR arm use tablet/iPad for any other use besides internet access?

11. The statement on line 291 says, “We found that 52% of the patients in the CDSS arm compared with only 39% in the SDR…..” However, according to figure, it is 39% and 23%, respectively.

12. On line 302, the authors argue that the negative correlation between overall satisfaction with the consultation and the length of the consultation was only evident in the CDSS arm where the GPs were using an iPAD. It is unclear if the GPs in SDR arm were also using iPad or not. In method section, the authors mention that both groups were provided with a tablet.

13. The last paragraph of the discussion section, the authors makes comments that are not the main objective of the study or supported by other studies. For instance, the authors say that CDSS can help achieve patient outcome and save money for healthcare system, but based on the most recent systemic review, there was no clear evidence that CDSS improves patient outcome or is cost effectiveness. (Black AD, Car J, Pagliari C, et al. The impact of eHealth on the quality and safety of health care: a systematic overview. PLoS Med. 2011;8(1):e1000387. Published 2011 Jan 18. doi:10.1371/journal.pmed.1000387

14. On line 335, the authors mention that the GPs in CDSS arm felt more confident with the diagnosis that they chose not to refer the patients. The questionnaire did not explore the GPs perspectives on how confident they felt, so we cannot jump to conclusion that they felt more confident based on the number of referrals placed. The reason for referral could be diagnostic, but also therapeutic.

15. The authors’ last comment on the discussion section (increased quality of life, happiness, increased productivity of the society) is too broad, and does not seem immediately relevant to the study objectives nor supported by the study.

16. The authors should include the limitation of the study (such as not being a randomized controlled trial, etc) to keep in mind the generalizability of the study.

6. PLOS authors have the option to publish the peer review history of their article (what does this mean?). If published, this will include your full peer review and any attached files.

Reviewer #1: No

---

## [Author Response · Author response to Decision Letter 0]

7 Jun 2020

Dear reviewer and editor,

We have in our rebuttal letter carefully responded to all your questions and comments. 

Yours sincerely,

Dr John Zibert

---

## [Editor Report · Decision Letter 1]

16 Jun 2020

Improved patient satisfaction and diagnostic accuracy in skin diseases with a Visual Clinical Decision Support System - a feasibility study with general practitioners

PONE-D-19-34156R1

Dear Dr. Zibert,

We’re pleased to inform you that your manuscript has been judged scientifically suitable for publication and will be formally accepted for publication once it meets all outstanding technical requirements.

Kind regards,

Elisa J. F. Houwink, MD, PhD

Academic Editor

PLOS ONE
---

## [Editor Report · Acceptance letter]

29 Jun 2020

PONE-D-19-34156R1 

Improved patient satisfaction and diagnostic accuracy in skin diseases with a Visual Clinical Decision Support System - a feasibility study with general practitioners 

Dear Dr. Zibert:

I'm pleased to inform you that your manuscript has been deemed suitable for publication in PLOS ONE. Congratulations! Your manuscript is now with our production department. 

Kind regards, 

on behalf of

Dr. Elisa J. F. Houwink 

Academic Editor

PLOS ONE